# Osteoblastogenesis Alters Small RNA Profiles in EVs Derived from Bone Marrow Stem Cells (BMSCs) and Adipose Stem Cells (ASCs)

**DOI:** 10.3390/biomedicines8100387

**Published:** 2020-09-28

**Authors:** Yan Yan, Clare Chang, Junyi Su, Morten T. Venø, Jørgen Kjems

**Affiliations:** 1Interdisciplinary Nanoscience Center, Aarhus University, 8000 Aarhus, Denmark; yanyan@inano.au.dk (Y.Y.); clarechangphd@gmail.com (C.C.); junyi@inano.au.dk (J.S.); morten.veno@omiics.com (M.T.V.); 2Omiics ApS, Åbogade 15, 8200 Aarhus N, Denmark; 3Department of Molecular Biology and Genetics, Aarhus University, 8000 Aarhus, Denmark

**Keywords:** multipotent stem cell, bone marrow stem cell, adipose stem cells, osteoblastogenesis, extracellular vesicles, microRNA, tRNA-derived small RNA

## Abstract

Multipotent stem cells (MSCs) are used in various therapeutic applications based on their paracrine secretion activity. Here, we set out to identify and characterize the paracrine factors released during osteoblastogenesis, with a special focus on small non-coding RNAs released in extracellular vesicles (EVs). Bone marrow stem cells (BMSCs) and adipose stem cells (ASCs) from healthy human donors were used as representatives of MSCs. We isolated EVs secreted before and after induction of osteoblastic differentiation and found that the EVs contained a specific subset of microRNAs (miRNAs) and tRNA-derived small RNAs (tsRNA) compared to their parental cells. Osteoblastic differentiation had a larger effect on the small RNA profile of BMSC-EVs relative to ASC-EVs. Our data showed that EVs from different MSC origin exhibited distinct expression profiles of small RNA profiles when undergoing osteoblastogenesis, a factor that should be taken into consideration for stem cell therapy.

## 1. Introduction

Multipotent stem cells (MSCs) are stem cells capable of self-renewal and differentiation into a variety of cell types, including osteoblasts, chondrocytes, and adipocytes, that are important for tissue regeneration and repair [1,2]. There are two mechanisms linked to the potential therapeutic function of MSCs: differentiation to target cells and the release of paracrine factors [3]. One subset of the paracrine factors secreted by MSCs are the exosomes–a class of extracellular vesicles (EVs) that are formed in the endosomal compartment and secreted when multivesicular bodies (MVBs) fuse with the plasma membrane [3,4]. EVs carry proteins and RNAs that can be taken up by recipient cells to allow cell-to-cell signaling in a paracrine manner [4,5,6,7].

The EVs target and enter recipient cells by receptor–ligand interactions, endocytosis, or by direct membrane-fusion [4,7]. In this way, the EVs can release their content, such as microRNAs (miRNAs) and tRNA-derived small RNAs (tsRNA), into the recipient cells and potentially affect gene expression [8,9,10]. MiRNAs are a class of small non-coding RNAs of ~22 nt in length that regulate gene expression at the post-transcriptional level by repressing translation or causing mRNA degradation [11,12]. In recent years, it was found that tRNAs are sometimes cleaved to generate two main kinds of small RNAs: tRNA halves (30–40 nts) and tRNA fragments (17–26 nts) [13,14,15]. The tRNA halves are generated by the stress-induced RNase angiogenin, which cleaves mature tRNAs at a specific site in the anticodon loop [16]. The tRNA fragments are formed when pre- and mature tRNAs are cleaved and trimmed by Dicer and RNase Z [17,18,19]. The tsRNAs function in two ways: tRNA halves can repress translation by displacing initiation factors from mRNA or binding to ribosomal components [20,21,22], while tRNA fragments can bind Argonaute proteins and repress gene expression as seen for miRNAs [14,23,24]. However, the functional implications of tsRNA regulation remains unclear.

In therapeutic bone regeneration, different strategies are used: direct injection of MSCs [25]; autologous bone grafting [26]; implantation of biomaterial or scaffolds together with MSCs and growth factors [27]. In stem cell therapy, bone marrow stem cells (BMSCs) have several advantages, such as high stability in culture and high ability to differentiate into osteoblasts [2]. However, BMSCs need to be extracted from bone marrow stroma, which is difficult to obtain, and the yield of BMSCs is generally low [2,28]. In contrast, adipose stem cells (ASCs) are more easily accessible and have higher yields [2,28]. Studies have shown that ASCs can differentiate into many different lineages, including osteoblastogenesis [28]; however, there is a high tendency for ASCs to spontaneously differentiate into adipocytes [2]. EVs derived from both BMSCs and ASCs have been shown to promote bone regeneration [3,29,30,31,32,33,34]. In addition, Baglio et al. found that tRNA-derived RNA fragments were enriched in BMSC- and ASC-exosomes and that the small RNA expression profile in the exosomes was dependent on MSC tissue origin [1]. However, this study did not address how osteoblastic differentiation affects the small RNA profile in MSC-derived exosomes.

The aim of the current study is to elucidate the small RNA profile in the EVs released from ASCs and BMSCs and the parental cells as they undergo in vitro differentiation to osteoblasts. Our data show that the expression of miRNAs and tRNA-derived small RNAs in MSCs and MSC-EVs changes during osteoblastogenesis and that the changes depend on cell origin.

## 2. Experimental Section

### 2.1. Cell Culture and Osteoblastic Differentiation

BMSCs from two donors were bought from American Type Culture Collection (ATCC, Manassas, VA, USA) and one donor was bought from Thermo Fisher Scientific (Waltham, MA, USA). ASCs from two donors were bought from Lonza (Basel, Switzerland) and one donor was bought from Stemmatters (Barco, Portugal). The characteristics of cell donors are provided in Appendix A. Cells were maintained in Dulbecco’s modified Eagle medium (DMEM) low glucose supplemented with 10% fetal bovine serum, and 1% penicillin and streptomycin at 5% CO_2_ and 37 °C.

Primary donor cells were differentiated between Passages 4 and 6. The cells were seeded at 70% confluency one day prior to differentiation and osteoblastic differentiation media contained high glucose DMEM supplemented with 10 nM dexamethasone (Sigma-Aldrich, St. Louis, MO, USA), 10 mM β-glycerophosphate (Sigma-Aldrich), 50 μM ascorbate-2-phosphate (Sigma-Aldrich), and 10 mM 1.25-vitamin D_3_ (Sigma-Aldrich). The differentiation media was prepared fresh on the day and changed every 2–3 days.

Alizarin red S (ARS) staining was performed to stain calcium deposit secreted by osteoblasts. Cells were collected at day 17 of osteoblastic differentiation (D17) and fixed with 70% ethanol at −20 °C for 1 h. Cells were PBS washed and stained with 40 mM Alizarin red S pH 4.2 (Sigma-Aldrich) for 20 min at room temperature after which the excess dye was washed with deionized water and PBS. The calcium deposits were analyzed using the Olympus IX71 Microscope (Tokyo, Japan).

### 2.2. EV Purification

Cells were seeded at 70% confluency in T175 flasks and incubated in TheraPEAK chemically defined mesenchymal stem cell media (Lonza) until they reached 100% confluency. EV-containing media was collected as D0 (no differentiation). The chemically modified media was used for EV collection to avoid the EV contamination from FBS. The same cells were subsequently differentiated with TheraPEAK containing 10 nM dexamethasone (Sigma-Aldrich), 10 mM β-glycerophosphate (Sigma-Aldrich), 50 μM ascorbate-2-phosphate (Sigma-Aldrich), and 10 mM 1.25-vitamin D_3_ (Sigma-Aldrich). The media was replaced every 2–3 days and the last media change at day 7 was collected as EV-containing media for D7. EVs were purified with a series of centrifugation steps at 4 °C: 10 min at 300× *g*, 20 min at 2000× *g*, and 30 min at 15,500× *g*. The supernatant was removed at the end of each centrifugation and used for the subsequent centrifugation step. Finally, EVs were pelleted at 100,000× *g* for 90 min at 4 °C (Beckman-Coulter, Optima L-80-XP ultracentrifuge, type 60Ti rotor, Brea, California, United States), the supernatant was removed and the pellet was resuspended in 300 μL PBS.

### 2.3. Nanoparticle Tracking Analysis (NTA) of EVs

The purified EVs were analyzed on a Nanosight LM10 device (Malvern Instruments, Malvern, UK) with a 405 nm laser. All measurements were captured five times with 60 s video captures with camera level set to 15 and detection threshold at 10. The data were analyzed with NTA software version 3.1 (Malvern Instruments) to obtain EV size distributions and concentrations.

### 2.4. Western Blotting of EVs

Cells were lysed with RIPA buffer (Invitrogen, Waltham, MA, USA) containing protease inhibitors (Complete Ultra Tablets, Roche, Basel, Switzerland). For the EVs, protease inhibitor was added after purification. Protein concentration was determined with a Bradford assay (BioRad, Hercules, CA, USA) and 25 μg protein was used. Proteins were prepared in 4× NuPAGE LDS sample buffer (Invitrogen) and loaded on a 4–12% NuPAGE Novex Bis-Tris gel (Invitrogen). Proteins were separated by gel electrophoresis and transferred onto a PVDF membrane. Immunoblotting was performed with antibodies against CD81 (sc-9158, Santa Cruz Biotechnology, Dallas, TX, USA), and Calnexin (ab10286, Abcam, Cambridge, UK), conjugated to HRP-conjugated goat anti-rabbit secondary antibody (DAKO, Agilent, Santa Clara, CA, United States) according to the manufactures’ protocols of antibodies. The membrane was visualized using SuperSignal West Femto maximum sensitivity substrate (Thermo Fisher Scientific).

### 2.5. RNA Extraction

Cell and EV RNA was extracted with miRNeasy mini kit (QIAGEN, Hilden, Germany) and eluted in 50 μL RNAse-free water. The concentration of cellular RNA was checked on a bioanalyzer RNA nano chip (Agilent, Santa Clara, CA, United States). The EV RNA was concentrated by ethanol precipitation by adding 50 μL RNAse-free water, 1 μL glycoblue (Ambion, Thermo Fisher Scientific), 10 μL 3M pH 5.5 sodium acitate (Ambion, Thermo Fisher Scientific), and 250 μL pre-cooled 99% ethanol. The samples were incubated at −20 °C overnight then centrifuged at 16,000× *g* for 20 min at 4 °C. The pellet was washed with 1 mL 75% ethanol, followed by centrifugation at 16,000× *g* at 4 °C for 5 min. The RNA pellet was resuspended in 7 μL RNase-free water.

### 2.6. Small RNA Library Preparation and Sequencing

Small RNA libraries were constructed with the Illumina TruSeq small RNA sample prep kit (Illumina, San Diego, CA, USA) using 1 μg (200 ng/μL) cellular RNA and 5 μL EV RNA as input. Since the RNA input in our EV samples was lower than standard protocols, the amount of adaptors was reduced to 1/10, the other reagents were halved, and the number of PCR cycles was increased from 12 to 15. While the library construction of cellular RNA was done according to the manufacturer’s protocol. Library fragments 140–160 bp in length were purified on a Pippin Prep machine (Sage Science, Beverly, MA, USA). The size and the purity of the cDNA libraries were validated on a bioanalyzer high sensitivity DNA chip (Agilent) and the concentration was determined using a KAPA library quantification kit (KAPA biosystems, Roche, Basel, Switzerland). The libraries were pooled and sequenced on Illumina HiSeq 2000 by Beijing Genomics Institute (BGI, Shenzhen, China).

### 2.7. Raw Data Processing

Raw data were quality filtered and trimmed by fastx toolkit, and adaptor sequences were removed using Cutadapt. Quality control was performed using FastQC to ensure high quality data. Filtered reads were first mapped to human tRNA sequences using Bowtie allowing 1 mismatch. Non-mapping reads were then mapped to miRNA sequences using Bowtie allowing 0 mismatches, though allowing addition of A and T nucleotides at the 3′end, since miRNAs often have untemplated A and U additions. Reads not mapping to miRNAs were mapped to other relevant transcriptomes (mRNA, rRNA, and other small RNAs) and then to the human genome. Expression analysis was done on miRNA and tRNA using DESeq2 in R. The miRNAs with raw mapped reads <50 in all the samples, including ASC D0, ASC D7, BMSC D0, BMSC D7, and their EVs, were considered to be low expressed and these miRNAs were not included in the differential expression analysis. The miRNAs with Abs(log2FoldChange) ≥ 1 and *p* value < 0.05 are deemed as significantly changed. The tsRNAs were determined to be significantly changed if the adjusted p-values were below 0.05. The accession numbers of the sequencing data are PRJNA661572.

For functional enrichment analysis for target genes of selected miRNAs, significantly differentially expressed miRNAs were uploaded to MIENTURNET (MicroRNA ENrichment TURned NETwork) [35], where validated miRNA target genes were found based on data from miRTarBase [36]. MIENTURNET allows analysis of multiple miRNAs to discover and visualize the collective targeting of the miRNA sets. Pathway analysis based on KEGG pathways was performed on the detected miRNA target genes and significant enrichment of individual pathways visualized in dotplots.

## 3. Results

### 3.1. Characterization of EVs from BMSCs and ASCs

Osteoblastic differentiation was induced in BMSCs and ASCs in culture and validated by Alizarin red S (ARS) staining for mineralization after 17 days (Figure 1A); mineralization is indicative of successful osteoblastic differentiation. The gene expression of *RUNX2*, *Osteoclacin*, *CollagenIA1*, and *ALP* were significantly upregulated at seven days of differentiation compared to non-differentiation in BMSCs (Appendix A). EVs were isolated from the medium collected at day zero (D0) or seven (D7) of osteoblastic differentiation and purified by ultracentrifugation. The EVs measured 120–170 nm in size and the yield is 2000–9600 vesicles/cell, as determined by nanoparticle tracking analysis (NTA) (Figure 1C). We also observed a small fraction of larger EV species, ranging from 200 to 600 nm that could include microvesicles or EV aggregates formed during ultracentrifugation [37]. In Western blot analysis, the EV marker CD81 was detected in both EVs and cells, while the endoplasmic reticulum-specific marker Calnexin (negative control) only showed up in cells (Figure 1B), confirming the purity of the EV fraction.

### 3.2. Small RNA Profiles of MSCs and MSC-EVs

We purified RNA from EVs and cells at D0 and D7 and used it for small RNA library construction and sequencing. Length distribution analysis revealed a single peak at 22 nts in cells corresponding to miRNAs, while the small RNA fraction in EVs appeared more heterogeneous with two main peaks at 22 nts (miRNAs) and 30–33 nts, which may correspond to RNA fragments derived from tRNAs (Figure 2A). According to the RNA sequencing profile, miRNAs were the predominant small RNA species in both types of cells, accounting for 56–63% of total mapped reads (Figure 2B,D). The EVs had very different small RNA profiles compared to their parental cells. Only 12% and 10% of the total mapped reads mapped to miRNAs in EVs from ASC and BMSC (ASC-EVs and BMSC-EVs, respectively) at D0, while the percentage of miRNAs in ASC-EVs at D7 was even lower (8%) (Figure 2C,E). In contrast, 28% of the mapped reads in BMSC-EVs at D7 corresponded to miRNAs (Figure 2C). The most abundant class of small RNAs found in the EVs was tsRNAs accounting for 47% and 52% of total mapped reads in ASC-EVs and BMSC-EVs, respectively, at D0 (Figure 2C,E). In the sequences from ASC-EVs at D7, 47% of total mapped reads were tRNA-derived, while there were only 14% tsRNAs in BMSC-EVs at D7 (Figure 2C,E).

Overall, our data show that osteoblastic differentiation has a stronger effect on the small RNA content of EVs in BMSCs compared to ASCs where only a very small change in the relative content of different RNA types was seen.

### 3.3. Osteoblastic Differentiation Alters miRNA Expression in MSCs and MSC-EVs

The miRNAs with <50 raw mapped reads in all samples, including ASC D0, ASC D7, BMSC D0, BMSC D7, and their EVs, were considered lowly expressed and therefore not included in the differential expression analysis. Using this cut-off rate, we detected 429 and 404 miRNAs in ASCs and BMSCs, respectively. For the EVs, the same cut-off rate gave 157 and 211 miRNAs in ASC-EVs and BMSC-EVs, respectively. A PCA analysis based on the miRNA profile showed that the MSCs and MSC-EVs clearly grouped separately (Figure 3A). We also found that ASCs and BMSCs have distinct miRNA profiles, suggesting lineage-specific expression patterns between the two MSC populations (Figure 3A). However, based on the miRNA profiles, both cells and EVs, surprisingly, grouped together more based on donor identity than on the osteoblastic differentiation stages (Figure 3A).

We next looked at how osteoblastic differentiation changed miRNA abundance in cells and EVs (Figure 3B). The changes in miRNA expression with Abs (log2FoldChange) ≥1 and *p*-value < 0.05 between D0 and D7 were deemed significant and miRNAs with Abs (log2FoldChange) ≥ 2 and *p* value < 0.05 marked in red in the volcano plot (Figure 3B). More differentially expressed miRNAs were identified in ASCs (57 miRNAs) compared to 35 miRNAs in BMSCs. Fewer miRNAs were differentially expressed in EVs: 4 miRNAs in ASC-EVs and 21 miRNAs in BMSC-EVs. Ten miRNAs were identified for both ASCs and BMSCs; therein, 3 miRNAs were consistently up-regulated and 5 miRNAs were consistently down-regulated in both ASCs and BMSCs, while 2 miRNAs (miR-146-5p and miR-222-5p) changed in opposite directions (Table 1): miR-146-5p was up in ASCs, but down in BMSCs and BMSC-EVs, while miR-222-5p was down in ASCs but up in BMSCs. Our data suggest that during osteoblastogenesis, ASCs and BMSCs share some miRNA pathways, but the main changes in the expression of miRNA were cell type specific.

Next, the differential expression of BMSC-EV miRNAs and how they compared with their parental cells were explored. We found 6 miRNAs to be significantly changed upon osteoblastogenesis in both BMSCs and BMSC-EVs: 2 up-regulated (miR-133a-3p and miR-483-5p) and 4 down-regulated (miR-146a-5p, miR-146b-5p, miR-155-5p, and miR-378a-3p) (Table 1). In addition, 15 miRNAs showed significant changes only in BMSC-EVs and 6 of these are abundant with baseMean >20,000 (Table 2), including 5 down-regulated miRNAs (miR-10a-5p, miR-10b-5p, miR-191-5p, miR-486-5p, and miR-100-5p) and 1 up-regulated miRNA (miR-22-3p) (Table 2).

In the ASC-EVs, only four miRNAs were differentially expressed during osteoblastogenesis: one up-regulated (miR-125b-2-3p) and three down-regulated (miR-769-5p, miR-425-5p, and miR-145-5p) (Table 3). MiR-125b-2-3p was also up-regulated in ASCs, while miR-425-5p was down-regulated in ASCs (Table 1). Compared to BMSC-EVs, miRNAs in ASC-EVs changed less after differentiation and no miRNAs were significantly changed in both ASC-EVs and BMSC-EVs.

We performed KEGG analysis on four groups of differentially expressed miRNAs: miRNAs identified in both BMSCs and ASCs, the ones identified in both BMSCs and BMSC-EVs, the ones identified in both ASCs and ASC-EVs, and the ones only identified in BMSC-EVs. However, there were only two miRNAs identified in ASCs and ASC-EVs and no pathways showed significance. In the analysis of miRNAs identified in both ASCs and BMSCs, there were five enriched pathways participating in osteoblastogenesis: HIF-1 signaling pathway, PI3K-Akt signaling pathway, parathyroid hormone synthesis, MAPK signaling pathway, and calcium signaling pathway (Figure 4A). It is interesting that HIF-1 signaling, PI3K-Akt signaling, and MAPK signaling were also enriched in the analysis of the miRNAs identified in BMSCs and BMSC-EVs (Figure 4B). The PI3K-Akt signaling pathway and MAPK signaling pathway were also enriched in the miRNAs only identified in BMSC-EVs, while p53 signaling pathway, signaling pathways regulating pluripotency of stem cells, cell cycles, and cellular senescence were only enriched in BMSC-EVs (Figure 4C).

### 3.4. Differential tsRNA Expression

The sequencing data for EVs showed a high proportion of reads mapping to tRNAs (Figure 2C,E). The three most abundant tsRNAs in ASCs, Gly-GCC, Val-AAC, and Val-CAC, accounted for 35%, 16%, and 16% of tsRNA reads, respectively (Table 4). In BMSCs, these three tsRNAs were also the most abundant, constituting 27%, 8%, 7% of the pool, respectively (Table 4). In ASC-EVs, 55% of tRNA reads mapped to Gly-GCC, 16% to Glu-CTC, and 16% to Gly-CCC and in BMCS-EVs, the three most abundant tsRNA were Val-CAC (43%), Val-AAC (33%), and Gly-GCC (13%) (Table 4). All of these abundant tsRNAs are 5‘ tRNA halves that are 30–33 nt in length (Figure 5).

Hierarchical clustering analysis was done based on the expression of tsRNAs and it showed that EVs exhibited distinct tRNA profiles compared to their parental cells and that ASCs and BMSCs were grouped separately (Figure 6A). It was also shown that BMSC-EVs from D0 and D7 grouped separately while ASC-EVs from both time-points grouped together based on donor identity (Figure 6A). We then looked at changes in expression for the individual tsRNAs using an FDR < 0.05 as a cut-off for significant changes in expression (marked in red in the volcano plot in Figure 6B–E). No tsRNAs were differentially expressed in ASCs and BMSCs undergoing differentiation, while 11 and 5 tsRNAs show significant changes in abundance in BMSC-EVs and ASC-EVs, respectively (Figure 6B–E, Table 5). Two tsRNAs were seen in both BMSC-EVs and ASC-EVs: Ile-AAT was up-regulated in both, while Phe-GAA was up-regulated in BMSC-EVs but down-regulated in ASC-EVs (Table 5). Among the 11 differentially expressed tsRNAs in BMSC-EVs, 8 are tRNA fragments that are shorter than 30 nt and 3 tsRNAs (Gly-CCC, Gly-GCC and His-GTG) are 5‘ tRNA halves (Table 5, Figure 5A,B). All three tRNA halves were down-regulated in BMSC-EVs undergoing osteoblastogenesis, while the other 8 tsRNA were all up-regulated (Table 5). In BMSC-EVs, the down-regulation of these three 5‘ tRNA halves led to the dramatic decrease of tRNA percentage seen in the total small RNA profile (from 52% to 14%) (Figure 2E).

## 4. Discussion

Osteoblastogenesis is a complex process that involves a number of changes in gene expression, cell morphology, and behavior. In this study, we have focused on the expression of small RNAs, mainly miRNAs and small RNA derived from tRNAs (tsRNAs), and their secretion into extracellular vesicles. We found that the subset of miRNAs and tsRNAs that are differentially regulated in EVs during differentiation are distinct from the cellular counterpart. As such, this suggests that small RNAs are actively being loaded into EVs rather than passively representing the profile present in the parental cells, which implies cargo loading as a passive process.

Most of the differentially expressed miRNAs during osteoblastic differentiation were cell type-specific. Only 10 miRNAs were found to be differentially expressed in both ASCs and BMSCs, and of these 2 miRNAs showed opposite regulation (miR-146-5p and miR-222-5p). MiR-146-5p was up-regulated in ASCs, but down-regulated in BMSCs and BMSC-EVs. Previous studies reported that miR-146b-5p was up-regulated during adipogenesis and negatively regulated Sirtuin 1 to promote adipogenesis, while it was down-regulated in chondrogenic differentiation of human bone marrow derived skeletal stem cells [38,39]. This result suggests that osteoblastogenesis is more pronounced in the BMSCs compared to the ASCs. In our data, miR-222-5p was down-regulated in ASCs but up-regulated in BMSCs. In a previous report from our lab, miR-222-3p was found to be down-regulated in osteoblastogenesis and anti-miR-222-3p promoted osteoblastogenesis [40]. MiR-222-5p was found to inhibit the differentiation from MSCs to smooth muscle cells by targeting *ROCK2* and α-smooth muscle actin [41]. However, there have been no data linking miR-222-5p to osteoblastic differentiation. Among the 10 miRNAs that changed in both ASCs and BMSCs upon osteoblastic differentiation, miR-210-3p and miR-335-3p were down-regulated and had relatively higher expression than the other down-regulated miRNAs in ASCs and BMSCs (Table 1). In our previous study, these two miRNAs were also down-regulated and acted as negative regulators of osteoblastogenesis [40] and other studies have supported a role for these two miRNAs in osteoblastogenesis [42,43]. Notably, most of the miRNAs with significantly differential expression were cell origin specific: 47 miRNAs were only differentially expressed in ASCs and 25 miRNAs only in BMSCs. Our data suggest that the origin of MSCs strongly influences the miRNA profile changes observed during osteoblastogenesis, which should be taken into consideration in stem cell application in bone regeneration.

Osteoblastic differentiation also affected the miRNA levels in EVs, especially the EVs derived from BMSCs. Six miRNAs show consistently differential expression in both BMSCs and BMSC-EVs: 2 were up-regulated (miR-133a-3p and miR-483-5p) and 4 down-regulated (miR-146a-5p, miR-146b-5p, miR-155-5p, and miR-378a-3p) (Table 1). A previous study found that miR-483-5p promoted osteoblastic differentiation and was involved in the development of osteoporosis [44]. In contrast to our study, miR-133a-3p was validated to be an anti-osteoblastic miRNA in some studies [45,46]. Consistent with our data, the 4 down-regulated miRNAs were also found to be anti-osteoblastic miRNAs in other studies [39,40,47,48]. We found 57 miRNAs to be differentially expressed in ASCs during osteoblastogenesis, but only 4 miRNAs in ASC-EVs. There was only one miRNA (miR-125b-2-3p) consistently up-regulated in both cells and EVs. However, there is no previous study linking the miRNAs of ASC-EVs to osteoblastogenesis, leaving their contribution to the process open. We noticed that the miRNA content in ASC-EVs changed less after differentiation than in the BMSC-EVs, which may indicate that ASC-EVs are less likely to play a role in paracrine signaling.

Unlike miRNAs, the differential expression of tsRNAs upon osteoblastogenesis was only detectable in MSC-EVs, but not in the parental cells. Here, the total amount of tsRNA reads decreased from 52% in BMSC-EVs D0 to 14% in BMSC-EVs D7, which may indicate a role of tsRNAs in the osteoblasic differentiation of BMSCs. Lee et al. found a significant amount of tsRNAs in the sequencing data of prostate cancer cell line and knock down of a tsRNA derived from the 3’ end of a Ser-TGA tRNA precursor impaired cell proliferation [13]. Kuscu et al. found that the expression of the 18 nts fragments derived from this tRNAs increased with the overexpression of the parental tRNAs and that the tRNA fragments repressed gene expression of mRNAs harboring complementary seed regions in a Dicer-independent and Ago-dependent manner [14]. Chiou et al. collected EVs from T cell culture and found that tRNA fragments were enriched upon T cell activation [9]. However, the precise function of the tsRNAs in osteoblastogenesis remains unclear.

In summary, our data showed that EVs derived from multipotent stem cells contained distinct miRNA and tsRNA profiles compared to the parental cells and characteristic alteration in their expression was seen during osteoblastic differentiation. What paracrine roles these EVs may play in the homeostasis of bodily functions remains to be studied further in the future.

## Figures and Tables

**Figure 1 biomedicines-08-00387-f001:**
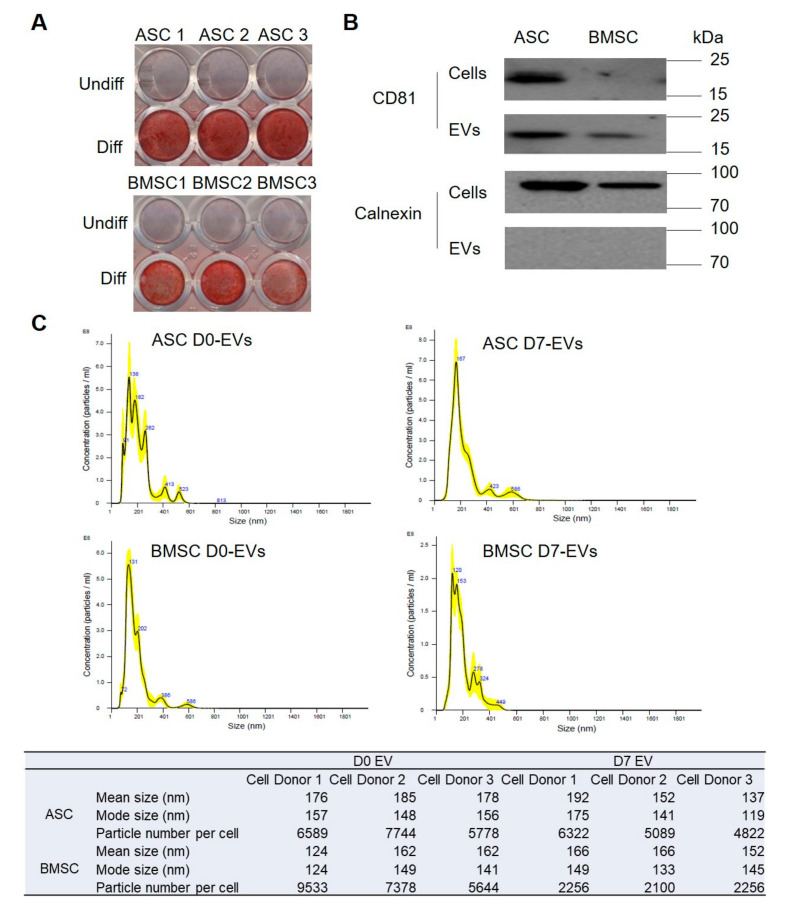
Characterization of extracellular vesicles (EVs) secreted from bone marrow stem cells (BMSCs) and adipose stem cells (ASCs). (**A**) Characterization of osteoblastic differentiation using Alizarin red S (ARS) staining. ASC1, ASC2, ASC3 present ASCs from three different donors; BMSC1, BMSC2, BMSC3 represent BMSCs from three different donors. (**B**) characterization of EV markers using Western blot. Cells and EVs are pools from three donors. (**C**) Size distribution of EVs based on nanoparticle tracking analysis (NTA). The NTA result is shown of one of the three donors and the table below shows the NTA results of all the three donors. D0 represents the time just before osteoblastic differentiation induced; D7 represents the day 7 of osteoblastic differentiation.

**Figure 2 biomedicines-08-00387-f002:**
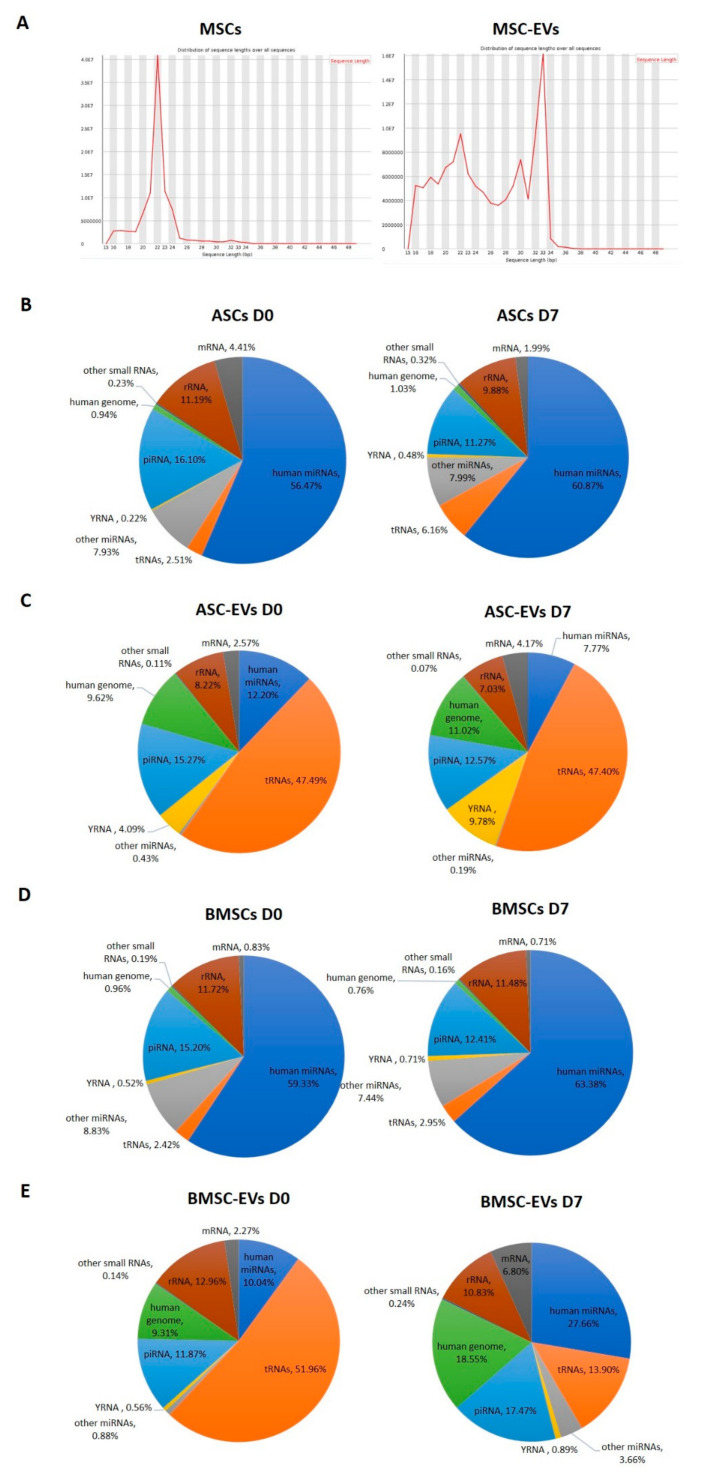
Overview of small RNA sequencing data in Multipotent stem cells (MSCs) and MSC-EVs. (**A**) Length distribution of total clean reads (clean reads mean the reads after adapter trimming and removal of low quality reads). Small RNA annotation in (**B**) BMSCs, (**C**) BMSC-EVs, (**D**) ASCs, and (**E**) ASC-EVs at day 0 (D0) and day 7 (D7) of osteoblastic differentiation.

**Figure 3 biomedicines-08-00387-f003:**
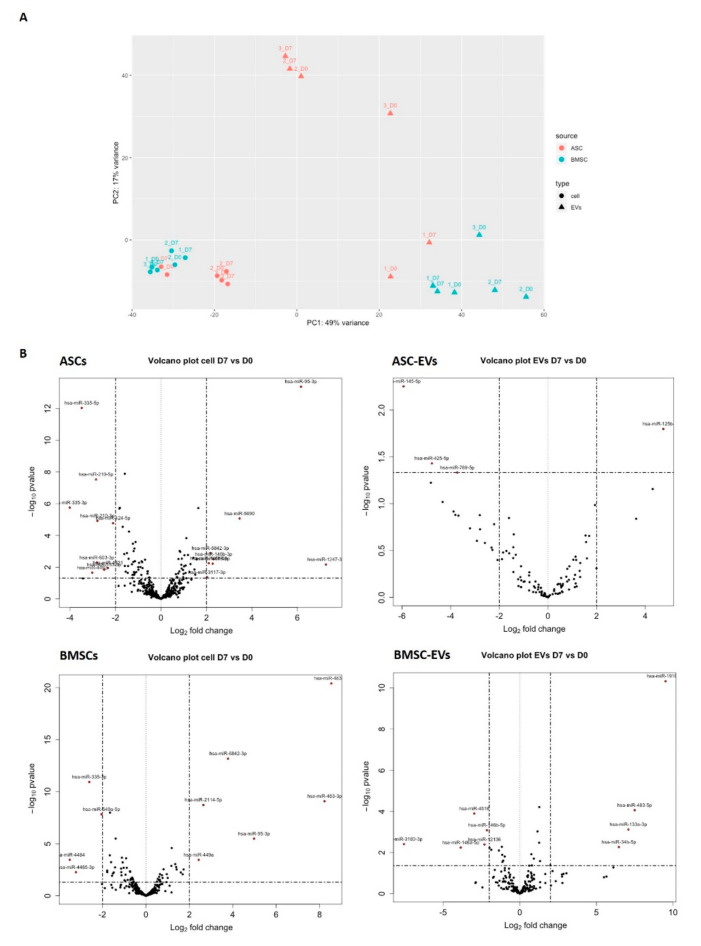
Differential expression analysis of miRNAs in MSCs and MSC-EVs in osteoblastic differentiation. (**A**) Principal component analysis **(**PCA) analysis (filled circle, cells; triangles, EVs; red, ASC; green, BMSC; numbers, different cell donors); (**B**) Differential expression analysis of miRNAs using DESeq2. The miRNAs with Abs (log2FoldChange) ≥ 2 and *p* value < 0.05 are deemed as significantly changed and marked in red in the volcano plot.

**Figure 4 biomedicines-08-00387-f004:**
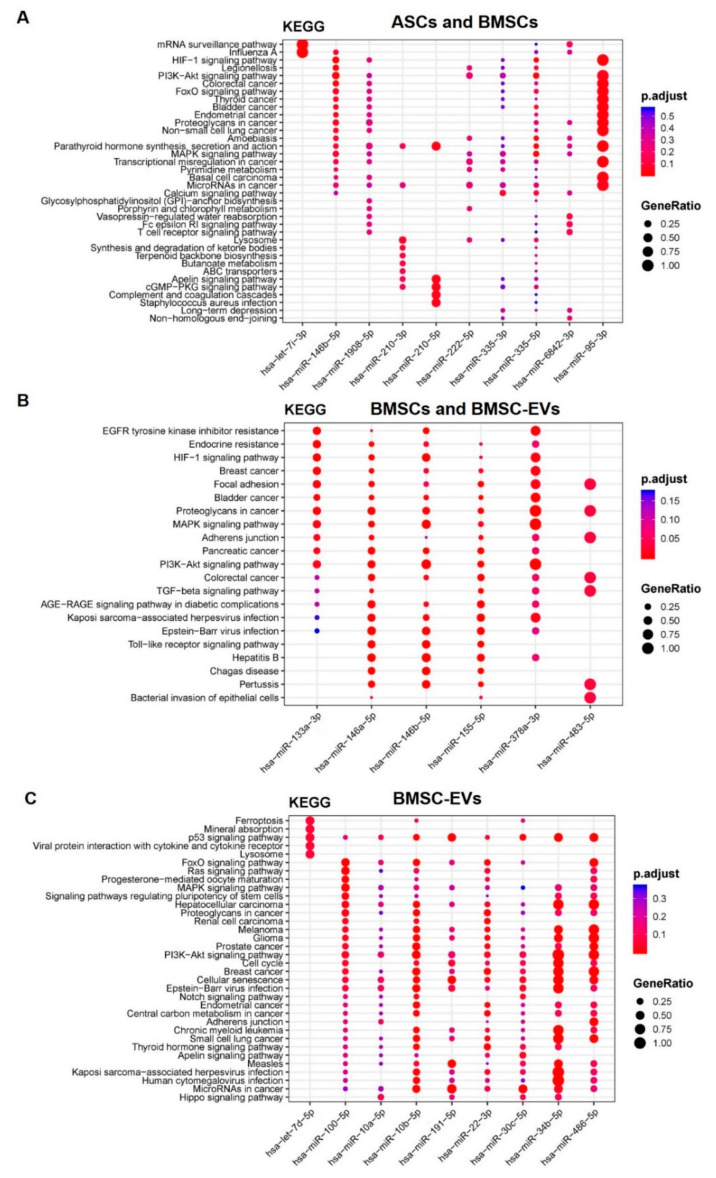
Functional enrichment analysis for target genes of differentially expressed miRNAs in osteoblastogenesis. (**A**) miRNAs identified in both ASCs and BMSCs. (**B**) miRNAs identified in both BMSCs and BMSC-EVs. (**C**) miRNAs identified only in BMSC-EVs. The *Y*-axis shows the annotation categories (e.g., KEGG pathways) and the *X*-axis shows the miRNAs. The color of the dots represents the adjusted *p* values (FDR) and the size of the dots represents gene ratio (the number of miRNA targets enriched in each category over the number of total genes associated to that category).

**Figure 5 biomedicines-08-00387-f005:**
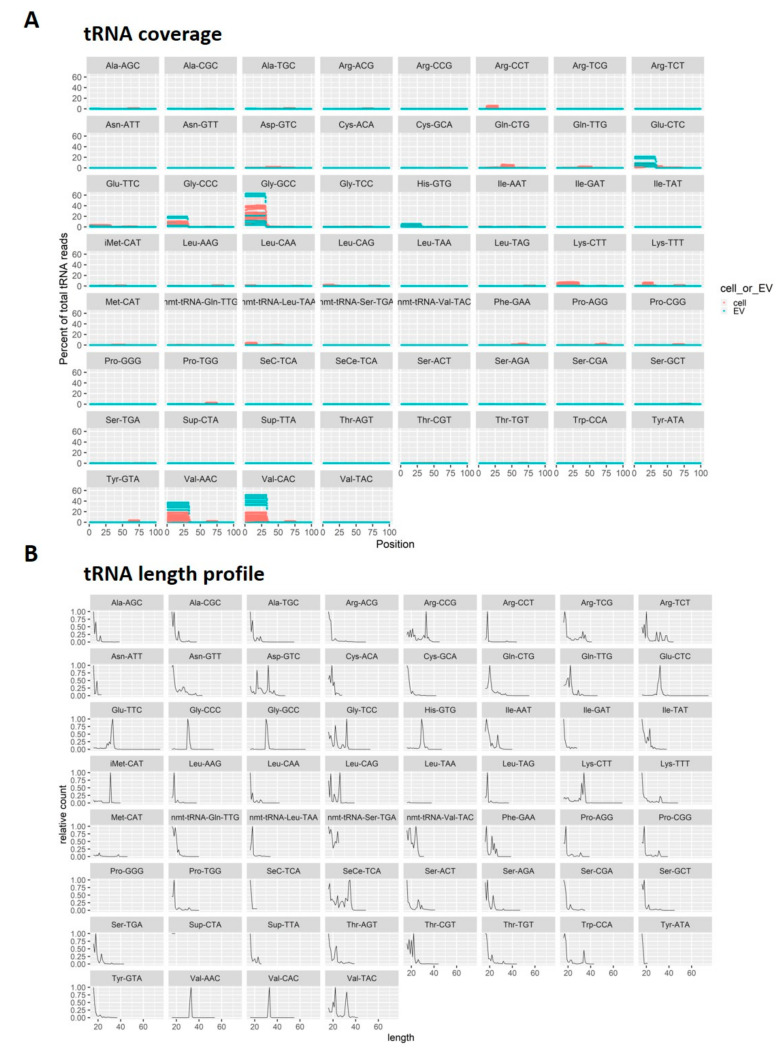
Analysis of small RNAs derived from tRNAs (tsRNAs). (**A**) The coverage position of tsRNAs on the mature tRNAs. (red, cells; green, EVs) (**B**) The length distribution of tsRNAs.

**Figure 6 biomedicines-08-00387-f006:**
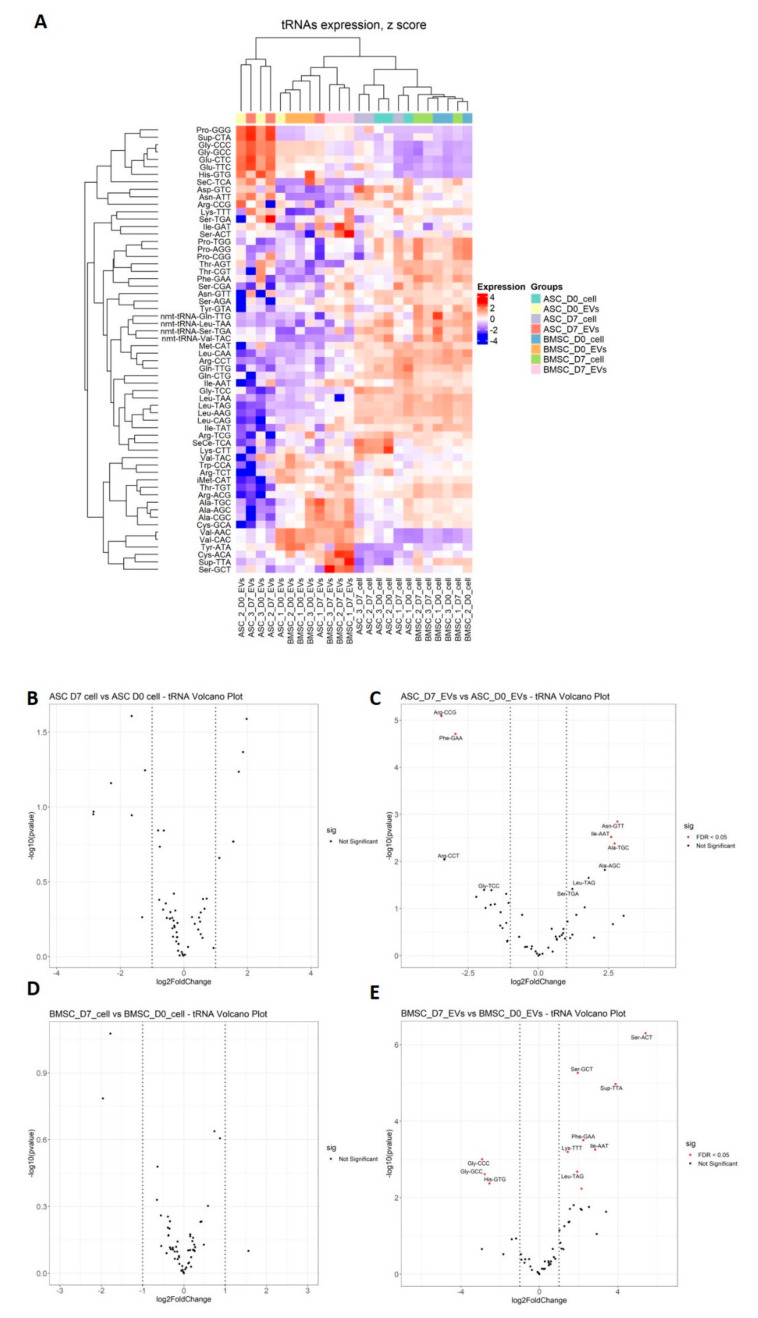
Differential expression analysis of tsRNAs in MSCs and MSC-EVs. (**A**) Heat map shows the expression profile of tsRNAs with unsupervised clustering of sample origin (**B**) Differential expression analysis of tsRNAs using DESeq2. The miRNAs with Abs (log2FoldChange) ≥ 1 and FDR < 0.05 are deemed as significantly changed and marked in red in the volcano plot.

**Table 1 biomedicines-08-00387-t001:** miRNAs showing differential expression upon osteoblastogenic differentiation (D7 vs. D0).

Sample	miRNA Name	BaseMean	Log2Fold Change	Sample	miRNA Name	BaseMean	Log2Fold Change
ASC cells	hsa-let-7i-3p	479	−1.6	BMSC cells	hsa-let-7i-3p	427	−1.0
	hsa-miR-146b-5p	20821	2.1		hsa-miR-146b-5p	3630	−1.1
	hsa-miR-1908-5p	39	1.5		hsa-miR-1908-5p	31	1.7
	hsa-miR-210-3p	1191	−2.8		hsa-miR-210-3p	1236	−1.7
	hsa-miR-210-5p	89	−2.9		hsa-miR-210-5p	104	−1.5
	hsa-miR-222-5p	151	−1.5		hsa-miR-222-5p	443	1.0
	hsa-miR-335-3p	1681	−4.0		hsa-miR-335-3p	3536	−2.6
	hsa-miR-335-5p	1248	−3.5		hsa-miR-335-5p	2835	−1.7
	hsa-miR-6842-3p	281	2.2		hsa-miR-6842-3p	65	3.8
	hsa-miR-95-3p	172	6.2		hsa-miR-95-3p	21	5.0
ASC cells	hsa-miR-125b-2-3p	808	1.1	ASC-EVs	hsa-miR-425-5p	15	−4.8
	hsa-miR-425-5p	1440	1.1		hsa-miR-125b-2-3p	29	4.7
BMSC cells	hsa-miR-133a-3p	120	1.3	BMSC-EVs	hsa-miR-133a-3p	15	7.1
	hsa-miR-146a-5p	86	−1.1		hsa-miR-146a-5p	26	−3.9
	hsa-miR-146b-5p	3630	−1.1		hsa-miR-146b-5p	1450	−2.2
	hsa-miR-155-5p	871	−1.4		hsa-miR-155-5p	183	−1.1
	hsa-miR-378a-3p	2747	−1.4		hsa-miR-378a-3p	1026	−1.9
	hsa-miR-483-5p	168	8.5		hsa-miR-483-5p	20	7.5

Note: The DeSEQ2 package was used to calculate baseMean, log2FoldChange, *p* value; miRNAs with Abs (log2FoldChange) ≥ 1 and *p* value < 0.05 were considered to have significantly differential expression and listed in this table. The miRNAs shown in the table had *p* value < 0.05.

**Table 2 biomedicines-08-00387-t002:** miRNAs with differential expression upon osteoblastogenic differentiation (D7 vs. D0) in BMSC-EVs.

miRNA Name	BaseMean	Log2FoldChange
hsa-miR-10b-5p	172,238	−1.1
hsa-miR-10a-5p	71,894	−2.0
hsa-miR-22-3p	44,127	1.3
hsa-miR-191-5p	30,963	−1.4
hsa-miR-486-5p	22,431	−1.2
hsa-miR-100-5p	20,117	−1.1
hsa-miR-146b-5p	1450	−2.2
hsa-miR-378a-3p	1026	−1.9
hsa-let-7d-5p	908	1.2
hsa-miR-30c-5p	370	1.3
hsa-miR-155-5p	183	−1.1
hsa-let-7d-3p	168	1.3
hsa-miR-12136	94	−2.3
hsa-miR-1910-5p	82	9.5
hsa-miR-4516	81	−3.0
hsa-miR-877-5p	42	2.0
hsa-miR-146a-5p	26	−3.9
hsa-miR-483-5p	20	7.5
hsa-miR-133a-3p	15	7.1
hsa-miR-3180-3p	14	−7.6
hsa-miR-34b-5p	10	6.5

Note: The DeSEQ2 package was used to calculate baseMean, log2FoldChange, *p* value and *p* adj; miRNAs with Abs (log2FoldChange) ≥1 and *p* value < 0.05 were considered to have significantly differential expression and listed in this table. The miRNAs shown in the table had *p* value < 0.05.

**Table 3 biomedicines-08-00387-t003:** miRNAs with differential expression upon osteoblastogenic differentiation (D7 vs. D0) in ASC-EVs.

miRNA Name	BaseMean	Log2FoldChange
hsa-miR-769-5p	20	−3.7
hsa-miR-425-5p	15	−4.8
hsa-miR-125b-2-3p	29	4.7
hsa-miR-145-5p	97	−5.9

Note: The DeSEQ2 package was used to calculate baseMean, log2FoldChange, *p* value and *p* adj; miRNAs with Abs (log2FoldChange) ≥1 and *p* value < 0.05 were considered to have significantly differential expression and listed in this table. The miRNAs shown in the table had *p* value < 0.05.

**Table 4 biomedicines-08-00387-t004:** Top 3 most abundant tRNAs in MSCs and MSC-EVs.

Sample	tRNAs	Percentage of Total Reads Mapped to tRNAs	Sample	tRNAs	Percentage of Total Reads Mapped to tRNAs
ASCs	Gly-GCC	34.59%	BMSCs	Gly-GCC	26.99%
	Val-AAC	15.97%		Val-CAC	8.46%
	Val-CAC	15.93%		Val-AAC	7.41%
ASC-Evs	Gly-GCC	54.59%	BMSC-Evs	Val-CAC	43.17%
	Glu-CTC	16.14%		Val-AAC	33.07%
	Gly-CCC	16.09%		Gly-GCC	13.07%

**Table 5 biomedicines-08-00387-t005:** tsRNAs with differential expression upon osteoblastic differentiation (D7 vs. D0).

Sample	tRNAs	BaseMean	Log2FoldChange	Note
BMSC-EVs	Ser-ACT	15	5.4	fragment
	Ser-GCT	753	2.0	fragment
	Sup-TTA	35	3.9	fragment
	Phe-GAA	826	2.2	fragment
	Ile-AAT	545	2.8	fragment
	Lys-TTT	4439	1.4	fragment
	Gly-CCC	236570	−2.9	halve
	Leu-TAG	393	1.9	fragment
	Gly-GCC	804325	−2.8	halve
	His-GTG	22356	−2.5	halve
	Thr-CGT	234	2.1	fragment
ASC-EVs	Arg-CCG	204	−3.5	halve
	Phe-GAA	826	−3.0	fragment
	Asn-GTT	147	2.8	fragment
	Ile-AAT	545	2.6	fragment
	Ala-TGC	1188	2.7	fragment

Note: The DeSEQ2 package was used to calculate baseMean, log2FoldChange, *p* value and *p* adj; tsRNAs with *p* adj < 0.05 were listed in this table. The miRNAs shown in the table had *p* adj < 0.05.

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
