# Peer review of "Osteoblastogenesis Alters Small RNA Profiles in EVs Derived from Bone Marrow Stem Cells (BMSCs) and Adipose Stem Cells (ASCs)"

_biomedicines, 2020, doi:10.3390/biomedicines8100387_

Round 1

Reviewer 1 Report

Thank you for the revision. The donors' data are interesting since one donor differs from sex and one other form ethnic group. Well done! It is a complete and very interesting paper.

Reviewer 2 Report

Dear Authors,

I am satisfied about revised version of manuscript and answers on the comments. In my opinion data relating to osteoblastic differentiation markers at different time selected for EVs isolation should improve the quality of the paper. 

This manuscript is a resubmission of an earlier submission. The following is a list of the peer review reports and author responses from that submission.

Round 1

Reviewer 1 Report

Title: Osteoblastogenesis alters small RNA profiles in EVs derived from bone marrow stem cells (BMSCs) and adipose stem cells (ASCs).

Yan Yan et al., in the manuscript entitled “Osteoblastogenesis alters small RNA profiles in EVs derived from bone marrow stem cells (BMSCs) and adipose stem cells (ASCs)” isolated EVs from bone marrow stem cells (BMSCs) and adipose stem cells (ASCs) from healthy human donors that were used as representatives of multipotent stem cells (MSCs). They isolated EVs secreted before and after induction of osteoblastic differentiation and found that the EVs from different MSCs origin exhibited distinct expression profiles of small RNA profiles when undergoing osteoblastogenesis, a factor that should be taken into consideration for stem cell therapy.

However, in my opinion, the work can be further improved by some suggestions:

  • The authors in Fig 1A must introduce the analysis of osteoblastic differentiation markers (osteocalcin, DMP-1, osteoadherin) to show data that support the eligibility of the experimental protocol used for osteoblastic differentiation.
  • Moreover, considering that osteoblastogenesis is a complex process that involves several changes in gene expression, cell morphology and behavior, the analysis of osteoblastic differentiation markers is necessary to support the hypothesis that osteoblastic differentiation has a stronger effect on EVs content. In my opinion the analysis of osteoblastic differentiation markers at different time selected for EVs isolation must be shown to better understand and support the data presented in this paper.
  • The authors must clarify if the data showed in Fig 1 and in subsequent experiments are data relating to one donor  or three donors.
  • In Fig 1B and 1C the authors should refer to the new guidelines of the scientific community, MISEV guidelines 2018 (https://www.ncbi.nlm.nih.gov/pubmed/30637094), and revise the text accordingly.
  • In concusion, this paper has a very descriptive impact but there is a lack of data related to a functional impact that could improve the quality of the manuscript.

Author Response

Dear Reviewer,

The authors are most grateful to the comments and advice. We appreciate the efforts invested in reviewing the manuscript. We have addressed all the comments below, and have changed the manuscript according to the comments where indicated. We believe that the result is a significantly improved manuscript.

  1. The authors in Fig 1A must introduce the analysis of osteoblastic differentiation markers (osteocalcin, DMP-1, osteoadherin) to show data that support the eligibility of the experimental protocol used for osteoblastic differentiation. Moreover, considering that osteoblastogenesis is a complex process that involves several changes in gene expression, cell morphology and behavior, the analysis of osteoblastic differentiation markers is necessary to support the hypothesis that osteoblastic differentiation has a stronger effect on EVs content. In my opinion the analysis of osteoblastic differentiation markers at different time selected for EVs isolation must be shown to better understand and support the data presented in this paper.

Answer: We agree with the reviewer´s comment. We did ARS staining on ASCs and BMSCs from all the donors at D17 (17 days of osteoblastic differentiation) and the mineralization is now shown at D17 for all the donors, which is included in Figure 1A. In addition, we quantified the gene expression of RUNX2, Osteocalcin, Collagen I A1 and ALP on the BMSCs from one donor at D0 and D7 (7 days of osteoblastic differentiation) and the results showed the upregulation of all four genes at D7 of osteoblastic differentiation. We have included the QPCR data in the supplementary data (supplementary figure 1) and revised the text (page 4 line 14-16). However, we are not able to provide differentiation markers for all cells preparations used for EV collection, because there is no RNA left.

  1. The authors must clarify if the data showed in Fig 1 and in subsequent experiments are data relating to one donor or three donors. In Fig 1B and 1C the authors should refer to the new guidelines of the scientific community, MISEV guidelines 2018 (https://www.ncbi.nlm.nih.gov/pubmed/30637094), and revise the text accordingly.

Answer: We thank the reviewer for the comment and we have now rephrased the text to clarify the results. The original figure 1A showed the results from one donor but we have now included the ARS staining results for all donors. The Western blot was done on pooled cells and EVs for each condition. Figure 1C shows the result for only one donor, but we did measure on EVs purified from all the donors. We summarized the NTA data in a table and included it in figure in Figure 1 C. The new MISEV guidelines recommend that western blots should include 1-3 EV markers and 1-3 non-EV markers. Accordingly, we have used CD81 as an EV marker and calnexin as a non-EV marker. The MISEV guidelines further recommend using NTA to quantify EVs and particle number secreted from one cell as one way to state the quantification of EVs. We calculated the EV numbers secreted per cell and put the results in the table in Figure 1 C. We have also revised the text accordingly (page 4 line 18 and 21, page 5 line 2-9).

  1. In conclusion, this paper has a very descriptive impact but there is a lack of data related to a functional impact that could improve the quality of the manuscript.

Answer: We thank the reviewer for the comment. We have now included a KEGG pathway analysis to predict the pathways targeted by the differentially expressed miRNAs (figure 4) and we revised the text accordingly (page 4 line 3-9; page 10, line 11-22; page 11 line1-7).

Reviewer 2 Report

Doctor Yan and colleagues presented the results concerning the analysis of the extracellular vesicles transcripts of stem cells induced to osteogenic differentiation. The paper is fine and the results interesting, even if some points need to be improved:

  1. Do the authors refer to multipotent stem cells, mesenchymal stem cells or to multipotent mesenchymal stromal cells? The minimum criteria of ISCT must be fulfilled in order to conclude that these are true MSCs. Please, provide all these details as supplementary materials.
  2. Please, specify the culture conditions in terms of temperature etc. Moreover, specify what "D17" means. Do the authors refer to "Day 17 of culture"? The same for D0.
  3. In Materials and Methods, the authors described the ars staining analysis at D17, while in results they reported "after 14 days". Which time is correct? Please, be consistent. Moreover, why did the authors choose this time? 
  4. The authors stained the sample by CD81, an exosomal marker, but they always referred to general "EVs". Did they focus only on exosomes sub-population? If yes, why did not they consider the use of exosomes isolation kits in order to increase the exosomes number?
  5. In the analysis, the authors exclude miRNAs <50 raw mapped reads. This passage is unclear. Did they find no vatiationin low abundant miRNA expression? I mean, did they exclude all the miRNA < 50 reads in both the considered time points or only in one? Did they observed any difference or oscillation in the quantity of those miRNA?
  6. In the Results section, page 7 line 25, the authors named "highly expressed" a group of 6 miRNA, but all but one are down-regulated. What did the author mean with the term "highly expressed"? The reader may be confused...
  7. The authors observed that miRNA form EVs grouped together based on donor identity. Please, report the general donor characteristics in order to understand the level of comparability: age, sex and ethnic group at least. Did the authors consider these variables?
  8. In the Discussion, authors reported other studies suggesting that osteoblastogenesis is more pronounced in BMSCs than in ASCs. However, they observed miR-222-5p down-regulated in ASCs and up-regulated in BMSCs. This miRNA down-regulates osteoblastogenesis. How do the authors justify this result?
  9. I strongly suggest the authors to include an analysis concerning the mRNAs/genes target of the highly differentially expressed miRNAs or tsRNAs. It willl help also the interpretation of the results.
  10. Please, provide the promised accession numbers  of the sequencing data.

Additional minor points:

  • Typo or grammar errors: page 2 line 4 "al"; page 3 line 9 "biffer" and line 10 "was".
  • Spacing errors: page 2 line 29; page 4 line 2; page 11 line 24.

Author Response

Dear Reviewer and Editor,

The authors are most grateful to the comments and advice. We appreciate the efforts invested in reviewing the manuscript and the concerns expressed. We have addressed all the comments below, and have changed the manuscript according to the reviewer`s comments where indicated. We believe that the result is a significantly improved manuscript.

1. Do the authors refer to multipotent stem cells, mesenchymal stem cells or to multipotent mesenchymal stromal cells? The minimum criteria of ISCT must be fulfilled in order to conclude that these are true MSCs. Please, provide all these details as supplementary materials.

Answer: We thank the reviewer for the comment. The cells we used in this study are mesenchymal stem cells. We bought the cells commercially (BMSCs were all from ATCC and ASCs were all from Stemmatters), and MSC validation was performed by the company (including the ISCT required tests), which is why we did not re-test them.

2. Please, specify the culture conditions in terms of temperature etc. Moreover, specify what "D17" means. Do the authors refer to "Day 17 of culture"? The same for D0.

Answer: We thank the reviewer for the comment and apologize for the unclarity. The cells were cultured at 5% CO2 and 37°C and we have specified the culture conditions in the manuscript (Page 2 line 26). D0 means just before adding the osteoblastic differentiation media and D17 means 17 days of osteoblastic differentiation. We have included an explanation of D0 and D17 in the manuscript (page 2 line 33 and 40).

3. In Materials and Methods, the authors described the ars staining analysis at D17, while in results they reported "after 14 days". Which time is correct? Please, be consistent. Moreover, why did the authors choose this time? 

Answer: We thank the reviewer for the comment and apologize for the mistake. It is D17 and we have corrected the results to say "after 17 days" instead of  "after 14 days" (page 4, line 13). Osteoblastogenesis and mineralization is a dynamic process and several time points can be chosen in the range from 12-20 days. We selected the day where the most mineralization occurred in this instance. The most important thing is to distinguish the difference in mineralization for induced and non-induced cells. Here, the control was grown alongside the osteogenic induced sample but in normal culture media.

4. The authors stained the sample by CD81, an exosomal marker, but they always referred to general "EVs". Did they focus only on exosomes sub-population? If yes, why did not they consider the use of exosomes isolation kits in order to increase the exosomes number?

Answer: We thank the reviewer for the comment and suggestion. In this study, we focus on all the extracellular vesicles pelleted at 100.000g, including exosomes.

5. In the analysis, the authors exclude miRNAs <50 raw mapped reads. This passage is unclear. Did they find no vatiationin low abundant miRNA expression? I mean, did they exclude all the miRNA < 50 reads in both the considered time points or only in one? Did they observed any difference or oscillation in the quantity of those miRNA?

Answer: We thank the reviewer for the comment and apologize for the unclear description. The sentence “The miRNAs with raw mapped reads < 50 in all the samples were considered to be lowly expressed and these miRNAs were not included in the differential expression analysis” refers to all the samples we included in this study, including ASC D0, ASC D7, BMSC D0, BMSC D7 and their corresponding EVs. If the miRNAs have >50 raw counts in only one sample, they were kept in the differential analysis. In the current study, we chose to focus on the miRNAs with robust expression. The low abundant miRNAs may also be interesting, but the data is too noisy to provide a clear conclusion. We changed the phrase “The miRNAs with raw mapped reads < 50 in all the samples were considered to be low expressed and these miRNAs were not included in the differential expression analysis” to “The miRNAs with raw mapped reads < 50 in all the samples, including ASC D0, ASC D7, BMSC D0, BMSC D7 and their corresponding EVs, were considered to be low expressed and these miRNAs were not included in the differential expression analysis” (page 3, line 44-45; page 6, line 22-23).

6. In the Results section, page 7 line 25, the authors named "highly expressed" a group of 6 miRNA, but all but one are down-regulated. What did the author mean with the term "highly expressed"? The reader may be confused...

Answer: We thank the reviewer for the comment and apologize for the unclear description. We have changed “15 miRNAs showed significant changes only in BMSC-EVs and 6 of these were classified as highly expressed (basemean ï¹¥20,000)” to “15 miRNAs showed significant changes only in BMSC-EVs and 6 of these are abundant with basemean ï¹¥20,000” (page 9, line 8-9).

7. The authors observed that miRNA form EVs grouped together based on donor identity. Please, report the general donor characteristics in order to understand the level of comparability: age, sex and ethnic group at least. Did the authors consider these variables?

Answer: We thank the reviewer for the comment and suggestion. Considering the sample size (n=3 for BMSC and n=3 for ASC), we did not make any correlations with donor characteristics. Instead, we focused on similarities of expressed RNAs in these cells based on differentiation states. In order to draw further conclusions based on donor characteristics, a larger sample size would garner better results.

8. In the Discussion, authors reported other studies suggesting that osteoblastogenesis is more pronounced in BMSCs than in ASCs. However, they observed miR-222-5p down-regulated in ASCs and up-regulated in BMSCs. This miRNA down-regulates osteoblastogenesis. How do the authors justify this result?

Answer: We thank the reviewer for the comment. We wrote on page 11 line 22-25 “In a previous report from our lab, miR-222-3p was found to be down-regulated in osteoblastogenesis and anti-miR-222-3p promoted osteoblastogenesis [38]. MiR-222-5p was found to inhibit the differentiation from MSCs to smooth muscle cells  by targeting ROCK2 and α-smooth muscle actin [39]. However, there has been no data linking miR-222-5p to osteoblastic differentiation.” So our previous worked showed that the partner strand, miR-222-3p, promotes osteoblastogenesis while the function of miR-222-5p (studied here) was not known.

9. I strongly suggest the authors to include an analysis concerning the mRNAs/genes target of the highly differentially expressed miRNAs or tsRNAs. It willl help also the interpretation of the results.

Answer: We thank the reviewer for the comment and suggestion. We have included a KEGG analysis of the differentially expressed miRNAs that show up in both BMSCs and ASCs, in both BMSCs and BMSC-EVs, in both ASCs and ASC-EVs, and the differential expressed miRNA only shown up in BMSC-EVs. The new results are shown in figure 4 and we have revised the text accordingly (page 4 line 3-9; page 10, line 11-22; page 11 line1-7).There is limited data available on the target of tsRNAs and it remains unclear how tsRNA function. We were therefore not able to run a similar target analysis for the tsRNAs.

10. Please, provide the promised accession numbers of the sequencing data.

Answer: We thank the reviewer for the comment. The accession number of the sequencing data is PRJNA661572.

Round 2

Reviewer 1 Report

Dear Authors,

I am satisfied about revised version of manuscript and answers on the comments. In my opinion data relating to osteoblastic differentiation markers at different time selected for EVs isolation should improve the quality of the paper. 

Reviewer 2 Report

The authors answered to all my questions and really improved the manuscript accordingly. Moreover, they added some necessary clarification. Well done.

I have one additional comment:

Concerning the donors' identity, the authors replied that the sample size is not sufficient in order to perform a statistical analysis.

I agree with the authors, in fact I did not ask a statistical analysis. They are reporting that the results grouped based on the donors. They must justify this evidence. Since a statistical analysis is not possible, they have to report the donors characteristics as I requested. The simple data regarding sex, age and ethnic group may be sufficient in order to understand if the differences in tsRNA and miRNA may be influenced by those elements or not. 

I consider this addition as Major revision because it is mandatory in order to support part of the manuscript's results.